# Optimization of Textile Waste Blends of Cotton and PET by Enzymatic Hydrolysis with Reusable Chemical Pretreatment

**DOI:** 10.3390/polym15081964

**Published:** 2023-04-21

**Authors:** Antika Boondaeng, Jureeporn Keabpimai, Preeyanuch Srichola, Pilanee Vaithanomsat, Chanaporn Trakunjae, Nanthavut Niyomvong

**Affiliations:** 1Kasetsart Agricultural and Agro-Industrial Product Improvement Institute, Kasetsart University, Bangkok 10900, Thailand; 2Department of Biology and Biotechnology, Faculty of Science and Technology, Nakhon Sawan Rajabhat University, Nakhon Sawan 60000, Thailand

**Keywords:** pretreatment, optimization, hydrolysis yield, textile waste, PET, cellulose

## Abstract

Textile waste usually ends up in landfills and causes environmental pollution. In this study, pretreatment methods for textile recycling, including autoclaving, freezing alkali/urea soaking, and alkaline pretreatment, were applied to textile waste with various cotton/polyester blending ratios. The best condition for enzymatic hydrolysis was a 60/40 textile waste blend of cotton/polyethylene terephthalate (PET) with a reusable chemical pretreatment (15% NaOH) at 121 °C for 15 min. The hydrolysis of pretreated textile waste by cellulase was optimized using response surface methodology (RSM) based on central composite design (CCD). The optimized conditions were 30 FPU/g of enzyme loading and 7% of substrate loading, which resulted in a maximum observed value of hydrolysis yield at 89.7%, corresponding to the predicted value of 87.8% after 96 h of incubation. The findings of this study suggest an optimistic solution for textile waste recycling.

## 1. Introduction

Textiles play an important role in daily life. They are generally made from two types of raw materials: (i) natural fibers derived from plants (cotton, coconut, pineapple, hemp, flax, and bamboo), animals (wool and silk), and minerals (asbestos); and (ii) synthetic fibers synthesized from petroleum-based resources (polyester, nylon, polyolefins, acrylics, and elastane) and regenerated from natural fibers (rayon) [1]. Textiles are ubiquitously used by humans and have a significant impact on the environment in terms of their production, use, and end of life.

Currently, fast fashion, which involves cheap, trendy, and mass-produced clothes, has a huge impact worldwide. The continuous growth of the fashion industry is harming the environment, especially by contributing to global warming. Approximately 51.4 million tons (MT) of textile fibers were produced worldwide in 2000. This number increased to 76 MT in 2010 and 108.3 MT in 2020 and is projected to reach 300 MT by 2050 [2,3]. Approximately 75% of textile waste ends up in landfills or is burned, whereas only 25% is recycled or reused [4]. China generates the most textile waste, approximately 20–26 MT annually, with a low utilization rate [5]. In 2017, only 3.5 MT of textile waste was recycled and reused in China, and the rest was sent to waste-to-energy incinerators [6]. In 2020, approximately 20% of 22 MT of textile waste was recycled and only 1.56 MT was converted into recycled fibers. They aimed to increase the recycling percentage to 25% by 2025 and 30% by 2030. Similarly, an annual 2 MT of recycled textile fibers produced by 2025 and 3 MT by 2030 are targeted [7]. In the United States, approximately 16 MT of textile waste is generated annually, only 15% of which is recycled or donated, and 85% ends up in landfills or is used in combustion; this constitutes the highest proportion of landfilled textile waste among the leading textile economies [8]. The estimated cost of sending textile waste to landfills is USD 45 per ton [9]; therefore, promoting recycling and recovery technologies for textile waste is important.

Cotton and polyester (polyethylene terephthalate, PET) are the most common natural and synthetic fibers, respectively, used in the textile industry. Generally, cotton-based textile waste contains 50–100% of cellulose—a proportion higher than that of any other lignocellulosic material [10,11,12]. Hence, it can serve as a low-cost renewable source of fermentable sugars to produce biological products via various pretreatments and subsequent biological processes. Owing to the highly ordered crystalline structure of cotton fibers, pretreatments, including milling and autoclave modifications and chemical pretreatments, have been deemed necessary to improve enzymatic hydrolysis [11,13,14,15].

PET, the most widely used polyester, is produced through a condensation reaction between terephthalic acid (TPA) and ethylene glycol (EG) derived from petroleum feedstock [16]. It is commonly utilized in various applications, such as food packaging, single-serve beverage bottles, textile, and clothing [16,17]. Although PET can be degraded by different chemical processes including glycolysis and methanolysis, only hydrolysis can break down the PET yield TPA and EG, the monomers used to generate new PET polymers [18,19].

To recycle textile waste, it is necessary to separate the cotton and PET components. One of the components should be depolymerized or degraded while the other component should be maintained. Alkaline pretreatment (NaOH) with the addition of a phase-transfer catalyst has been reported to be effective for depolymerizing polyethylene terephthalate (PET) at temperatures of 70–95 °C and alkalinity in the range of 5–15% NaOH [20,21,22,23]. PET was hydrolyzed at an ester bond with the formation of a disodium terephthalic salt and ethylene glycol (EG) in the liquid phase. Three products were obtained from the process: cotton cellulose, TPA, and an aqueous phase containing EG. Terephthalic acid (TPA) can be precipitated by adjusting the aqueous phase to pH 2–3 [24]. To recycle cotton cellulose, it can be degraded with different solvents [13] or with enzymes produced by microorganisms [25]. The degrading of cotton cellulose can be achieved using cellulase enzymes including endoglucanases (EC 3.2.1.4), exoglucanases (EC 3.2.1.74), and β-glucosidases (EC 3.2.1.21) [26]. Cellulases are used in various industries and applications, including food, paper and pulp, textiles, pharmaceuticals, alcoholic beverages, starch processing, biofuel, and production [27,28]. Most commercial cellulases are derived from fungi, especially *Trichoderma* and *Aspergillus* species [29].

This study explores the possibility of cotton-based textile waste hydrolysis, which was pretreated using different methods. The separation of TPA and EG were not considered in this study. The optimal conditions for regenerated cellulose hydrolysis were evaluated using a statistical method. The key factors influencing enzymatic hydrolysis were investigated: enzyme and substrate loading.

## 2. Materials and Methods

### 2.1. Textile Waste Pretreatment

Different textile waste blends of cotton and PET 35/65 and 60/40, obtained from Yong Udom Karn Tho Co., Ltd. (Bangkok, Thailand) and L.V.W. Group Co., Ltd. (Nakhon Pathom, Thailand), respectively, were used as feedstocks in this study, and pure cotton obtained from L.I.S. International Co., Ltd. (Bangkok, Thailand) was used as a positive control. All types of textile waste were cut into small pieces (approximately 0.5 × 0.5 cm^2^) and pretreated using three different modification methods: autoclaving, freezing alkali/urea (KemAus, Cherrybrook, Australia) soaking, and alkaline pretreatment (NaOH, KemAus, Cherrybrook, Australia). Crude textile without pre-treatment was used as a control. For the autoclave pretreatment, the textile waste was autoclaved at 121 °C for 15 min after adding the mineral solution to adjust the desired initial moisture content. For freezing alkali/urea soaking, NaOH (7% *w*/*v*) and urea (12% *w*/*v*) were added to textile waste, and the mixture was frozen at −20 °C for 6 h [15]. The chemical residues were removed by washing with deionized water. The textile samples were dried to a constant weight in an oven at 40 °C. Finally, for alkaline pretreatment, the textile waste was soaked for 3 h in NaOH solution (15%) at 80 °C or autoclaved at 121 °C for 15 min [30]. The pretreated textile samples were washed with tap water and dried overnight in an oven prior to use. The recovery of cellulose in the pretreatment was calculated using the following equation (Equation (1)) [14]:(1)$$\mathrm{Recovery}\;\mathrm{of}\;\mathrm{cellulose}=\frac{W_{r}}{W_{c}}\;\times \;100\%$$
where $W_{r}$ is the weight of regenerated cellulose from the pretreatment and $W_{c}$ is the weight of cellulose in the employed waste textiles for pretreatment.

### 2.2. Enzymatic Hydrolysis

Regenerated cellulose from textile waste blends of cotton/PET (35/65 and 60/40) was subjected to enzymatic hydrolysis. Pretreated pure cotton was used as the positive control. Commercial cellulase (Cellic CTec2, 185 FPU/mL; Novozyme A/S, Basgsværd, Denmark) was used in this study. Regenerated cellulose (2 g) was added to a citric buffer (100 mL, 50 mM, pH 4.8) at an enzyme dosage of 25 FPU/g substrate. Hydrolysis was performed at 50 °C at 200 rpm for 96 h. Samples were withdrawn periodically to determine the hydrolysis yield using Equation (2) [15]:(2)$$\mathrm{Hydrolysis}\;\mathrm{yield}\;(\%)=\frac{\mathrm{Amount}\;\mathrm{of}\;\mathrm{glucose}\;\mathrm{released}\;(g)}{\mathrm{Amount}\;\mathrm{of}\;\mathrm{initial}\;\mathrm{glucose}\;\mathrm{in}\;\mathrm{substrate}\;(g)\times 1.111}\times \;100\%$$

The amount of glucose was measured using a high-performance liquid chromatography column (HPLC, Shimadzu LC-20A, Kyoto, Japan) connected to the Aminex HPX-87P column and refractive index detector (RID) using deionized water as the mobile phase under the following conditions: flow rate, 0.6 mL/min; column temperature, 80 °C [31].

### 2.3. Optimization of Pretreated Textile Waste Hydrolysis

The effect of the test variables on the enzymatic hydrolysis of pretreated textile waste was studied using response surface methodology (RSM). The central composite design (CCD) was used to evaluate the relationship between experimental factors and observed results. The investigated factors were enzyme loading (17.9 FPU/g substrate to 32.07 FPU/g substrate) and substrate loading (2.17–7.83%), with each at five different levels based on the factorial design at two levels. The regenerated cellulose from textile waste, pretreated with 15% NaOH at 121 °C for 15 min, was used as substrate. Each sample was slurried in 100 mL of a citrate buffer (50 mM, pH 4.8) and the commercial cellulase was added to initiate the hydrolysis, which was conducted at 50 °C and 200 rpm for 96 h. The CCD consisted of 2^2^ factorial points with four star points (α = ±1.41) and three replicates at the center point. A total of eleven experiments were performed to optimize the parameters. Equation (3) was fitted to evaluate the effect of each independent variable on the response.
(3)$$Y=a_{0}+a_{1}X_{1}+a_{2}X_{2}+a_{12}X_{1}X_{2}+a_{11}X_{1}^{2}+a_{22}X_{2}^{2}$$
where Y is the predicted response (hydrolysis yield, %), $a_{0}$ is a constant term, $a_{1}$ and $a_{2}$ are linear terms, $a_{11}$ and $a_{22}$ are quadratic terms, $a_{12}$ is an interaction term, and $X_{1}$ and $X_{2}$ are the test variables studied.

All experiments were conducted in triplicate with three replicates of the center points to verify the accuracy of the model predicted using Design Expert software (Stat Ease, Version 7.0, Minneapolis, MN, USA). Subsequently, a validation experiment was performed to verify the predicted values obtained from software analysis.

### 2.4. Effect of Incubation Time for Pretreated Textile Waste Hydrolysis

The enzyme was incubated with pretreated textile waste as a substrate to determine the effect of the incubation time on enzymatic hydrolysis. Samples were withdrawn at 24 h intervals for 168 h, and hydrolysis yield was measured as previously described.

### 2.5. SEM Analysis of Textile Waste Substrate

The physical changes in the pretreated textile substrates were observed using scanning electron microscopy (SEM). The textile surface was imaged before and after SSF at a magnification of 3000× and a voltage of 5 kV using a field-emission SEM (SU8020, Hitachi, Tokyo, Japan).

### 2.6. Fourier Transform Infrared (FTIR) Analysis

An FTIR spectrometer (Nicolet IR200 FTIR, Thermo Scientific, Madison, WI, USA) was used to analyze the textile samples before and after pretreatment. The spectra were acquired over the range of 400–4000 cm^−1^ with a spectral resolution of 4 cm^−1^ and plotted as intensity versus wavenumber [32]. Each evaluated spectrum is the mean of 32 scans.

### 2.7. Statistical Analysis

To determine the differences between treatment means, a statistical analysis of variance was performed, followed by Duncan’s multiple range test, using SPSS Software v. 20.0 (IBM Analytics, New York, NY, USA). Values were considered significant when *p* < 0.05.

## 3. Results and Discussion

### 3.1. Sugar Recovery from Textile Waste after Pretreatment

Textile waste composes of the crystalline structure of cellulose in cotton fibers and PET, which hinders enzymatic hydrolysis. Therefore, various pretreatment or modification methods have been proposed to improve the accessibility of enzymes and enhance hydrolysis yield. We subjected textile waste to three different pretreatment methods: (i) autoclaving, (ii) freezing alkaline/urea, and (iii) NaOH pretreatment (15% *w*/*v* and 80 °C, 3 h or 121 °C, 15 min). After pretreatment, the crude and pretreated textile waste samples were subjected to enzymatic hydrolysis for 96 h at 2% of substrate loading and an enzyme dosage of 25 FPU/g of commercial cellulase with carboxymethyl cellulase (CMCase) and β-glucosidase at 720.95 U/g and 114,663.89 U/g, respectively. Figure 1 shows the hydrolysis yields of textile fibers using commercial cellulase under various pretreatment conditions.

Autoclaving is a widely used pretreatment or modification technique applied for feedstock preparation. In general, the autoclave technique affects the morphology of the materials [15]. In this study, the results indicated that the hydrolysis yield from a 35/65, 60/40 textile waste blend of cotton/PET and pure cotton increased from 3.83 ± 1.01%, 4.12 ± 1.12%, and 5.50 ± 0.25% to 6.63 ± 2.90%, 9.79 ± 1.02%, and 16.78 ± 1.06%, respectively, with materials autoclaved prior to enzymatic hydrolysis. The results showed a positive correlation between the hydrolysis yield and cotton content. This may be attributed to the mild hydrothermal treatment in an autoclave (121 °C, 15 psi), which partially damaged the material during the pressurized steaming process, and thus provided a larger contact area for enzymatic hydrolysis [33].

Freezing NaOH/urea pretreatment has been reported as a new cellulose pretreatment solvent with low cost and no pollution [34]. As shown in Figure 1, the hydrolysis yield from a 35/65 and 60/40 textile waste blend of cotton/PET, as well as pure cotton increased from 3.83 ± 1.01%, 4.12 ± 1.12%, and 5.50 ± 0.25% to 20.95 ± 2.17%, 48.72 ± 1.46%, and 34 ± 0.45%, respectively. The results show that the hydrolysis yield with materials treated with freezing NaOH/urea was higher than that from autoclaved materials. The alkali hydrate can modify the structure of textile waste by depolymerizing PET, penetrating the amorphous region of cellulose, causing cellulose swell, and reducing its crystallinity, [24,34]. Additionally, the addition of urea components can bind free water and prevent cellulose chain interactions through hydrogen bonds [35].

Alkaline pretreatment combined with a hydrothermal process is a modified technique used in this study. Pretreatment with 15% NaOH at 121 °C for 15 min increased the maximum hydrolysis yields of cotton/PET mixtures in the proportions of 35/65, 60/40, and jeans (positive control) from 3.83 ± 1.01%, 4.12 ± 1.12%, and 5.50 ± 0.25% to 58.12 ± 1.32%, 66.74 ± 2.72%, and 56.67 ± 8.79%, respectively, which is higher than that from materials pretreated with 15% NaOH at 80 °C for 7 h. NaOH pretreatment at 121 °C for 15 min was performed at concentrations ranging from 5–15% *w*/*v*. The highest hydrolysis yield was obtained from a 60/40 textile waste blend of cotton/PET pretreated with 15% NaOH at 121 °C for 15 min, which was used in further experiments. As mentioned above, the autoclaving mechanism can break down the partial structure of materials, making cotton-based textile waste more accessible for alkaline solution hydrolysis. Therefore, the alkaline solution can easily depolymerize PET and reduce cellulose crystallinity. In addition, when enzymatic hydrolysis was performed on the 60/40 textile waste blend of cotton/PET with five reusable NaOH pretreatments, the hydrolysis yields of the pretreated textile waste were similar. This result implies that the pretreatment method can modify the structure and facilitate the reduction of the crystallinity of cellulose fiber and polyester in textile waste and enhance its susceptibility to subsequent hydrolysis.

### 3.2. Change of Surface Morphology and FTIR Investigation

Figure 2 shows scanning electron micrographs (SEM) of the surface of this textile waste blend before (Figure 2a) and after pretreatment with reusable chemicals (Figure 2b–f) at a magnification of 3000×. As shown in Figure 2, a crude textile with a slightly rough and uneven surface was clearly identified. This micrograph was similar to that observed in the study by Shen et al. [14]. After pretreatment with 15% NaOH at 121 °C for 15 min, the shape of textile waste changed (Figure 2b). The structure of the crude textile was partially broken into rugged fibers because of the digestion of cotton/PET by NaOH. The PET was hydrolyzed, and sodium terephthalate was released while cotton fibers only swelled by the penetration of the amorphous area and the disruption of the neighboring crystalline regions of cellulose by NaOH [36,37]. To check the recyclability and the effect of the NaOH solution on the structure of this textile waste blend, pretreatment with the same solution was further examined and detected by SEM. The economics and reduction of chemical waste are the main factors to consider in the recycling process. To determine whether the aqueous NaOH can be reused and the effect of this reused NaOH on product quality, the same solution was used to treat textile waste blend samples five times in a row. As shown in Figure 2b–f, all cycles achieved almost completely degraded PET and swelled cotton fibers. The structure of textile waste (Figure 2c–f) showed partially broken into rugged fibers similar to the structure in the first cycle (Figure 2b). The cellulose content was measured in each cycle and remained constant (93.47–94.78%). This result implies that reusable chemicals can maintain the destructive properties of textile waste.

FTIR spectra were obtained to confirm the characteristics of the polymer within the cotton/PET textile waste blend before and after pretreatment (Figure 3). The cotton/PET textile waste blend showed the corresponding infrared absorption bands of both PET and cellulose polymers. The absorption bands of the PET polymer were detected at 3440 cm^−1^ (O-H stretching), 1706 and 1060 cm^−1^ (C=O, stretching), 1412 and 723 cm^−1^ (C-H stretching), and 1342 cm^−1^ (C-O stretching) [38]. The cellulose polymer presented bands in the regions at 3334 cm^−1^ (-OH, stretching), 2898 cm^−1^ (C-H, stretching), 1642 cm^−1^ (-OH, bending), 1429 cm^−1^ (C-H, bending of CH_2_), 1162 and 1028 cm^−1^ (C-O, stretching), and 895 cm^−1^ (C-O-C, bond) [32,39]. Analysis of the FTIR spectra of pretreated textile waste showed the most bands in the cellulose polymer regions. This result suggests that the cellulose content of textile waste after pretreatment with 15% NaOH at 121 °C for 15 min was slightly high, with a yield of 94.2 ± 0.57%.

### 3.3. Optimization of Pretreated Textile Waste Hydrolysis

Response surface methodology (RSM) based on central composite design (CCD) was used to determine the optimal concentration of two independent variables, enzyme loading ($X_{1}$) and substrate loading ($X_{2}$), which affected the hydrolysis yield. The average of the three replicate values of hydrolysis yield is shown in Table 1, and the results of the regression analysis are presented in Table 2. The regression-based determination coefficient, R^2^, was evaluated to test the fit of the model equation. The value of the determination coefficient (R^2^ = 96.8) could explain 96.8% of the fit between the developed model and experimental data, and the remaining 3.2% was affected by other variables. The ANOVA of the regression model demonstrated that the model was statistically significant because of Fisher’s F-test (30.14) and a very low probability value (*P*-model > F = 0.001). The *p*-value was 0.0011, and a *p*-value < 0.0500 suggests important model terms. Values > 0.1000 suggest that the model terms were insignificant. The 2.11 lack-of-fit F-value means that the statistical value relative to the pure error was not significant, which is good for the model. This result indicates that the response equation provides a suitable model for the relationship between the independent variables and the response. The hydrolysis yield can be predicted using Equation (4):(4)$$Y=56.78+13.05X_{1}+11.93X_{2}+0.17X_{1}X_{2}+5.8X_{12}+0.08X_{22}$$
where Y is the hydrolysis yield (%), $X_{1}$ is the enzyme loading (FPU/g substrate), and $X_{2}$ is the solids loading (%).

The model coefficients tested by the regression analysis for each variable are listed in Table 2. The results revealed that enzyme loading ($X_{2}$) and substrate loading ($X_{2}$) had significant effects on hydrolysis yield. The quadratic term for enzyme loading ($X_{1}^{2}$) was significant (Table 2). The interaction between enzyme loading and substrate loading ($X_{1}X_{2}$) and the quadratic term of substrate loading ($X_{2}^{2}$) were > 0.05, indicating the non-significance of these coefficients.

The interaction relationships and optimal values of the variables were determined by examining the response surface plots of the independent factors on the hydrolysis yield, which are shown in Figure 4. No significant interaction was observed between enzyme loading and substrate loading for the hydrolysis yield, as shown by the high *p*-value (0.9559 > 0.05) in Table 2. The hydrolysis yield increased with increasing enzyme loading and substrate loading concentrations. A maximum hydrolysis yield of 87.8% was obtained. Figure 3 shows that the hydrolysis yield increased with an increase in both enzyme loading and substrate loading. This may be because an increase in enzyme loading results in increased cellulose digestibility. The hydrolysis yield rose as the enzyme loading and substrate loading increased up to 20 FPU/g to 30 FPU/g and 3% to 7%, respectively, but declined when they went beyond that range.

The optimized values of the variables were obtained from the regression equation and the response surface contour plots using Design Expert software. The model predicted that the maximum hydrolysis yield of 87.8% occurred at 30 FPU/g of enzyme loading and 7% of substrate loading. Validation of the experimental model, repeated three times under the optimum conditions, showed that the observed value of 89.7% was close to the predicted value of 87.8% after 96 h of incubation. The hydrolysis yield of 89.7% was approximately 1.6-fold higher than that before optimization. During hydrolysis, the cellulose component decomposed into soluble sugars (i.e., glucose) and separated from the solid residue (i.e., PET) by filtration at the end of hydrolysis. The time profile of hydrolysis yield is shown in Figure 5. The maximum hydrolysis yield of 100% was observed after 144 h of incubation.

Few studies have reported the degradation of textile waste. Hu et al. [15] reported the feasibility of using textile waste as a feedstock for cellulase production through solid-state fermentation by *Aspergillus niger* CKB. After 168 h of cultivation on an autoclaved textile blend of cotton/polyester of 80/20, the highest cellulase activity of 1.18 FPU/g with CMCase, β-glucosidase, and avicelase activities of 12.19 U/g, 1731 U/g, and 2.58 U/g, respectively, was achieved. Fungal cellulase was applied for textile waste hydrolysis, resulting in a recovery yield of 70.2%. Shen et al. [14] studied the recovery of polyester and fermentable sugars from cotton-based textile waste using phosphoric acid pre-treatment. Under optimized conditions (85% phosphoric acid, 50 °C, and 7 h), 100% polyester recovery with a maximum sugar recovery of 79.2% by enzymatic hydrolysis was achieved. As reviewed by Li et al. [40], textile waste pretreated by NaOH/urea significantly increased the hydrolysis yield compared with untreated textile waste. The glucose recovery yield reached the maximum at cellulase of 20 FPU/g, β-glucosidase of 10 U/g, and substrate loading of 3% at 50 °C and pH 5. After 96 h hydrolysis, the maximum glucose recovery yield of 98.3% was achieved.

This study addressed the challenge of managing textile waste. Recycling textile waste using biotechnology is a promising strategy to achieve sustainability and reduce the environmental impact of the textile industry. Cotton-based textiles are usually composed of polyester and cotton blended in different ratios, and their crystalline structures can obstruct enzymatic hydrolysis. Pretreatment technologies involve mechanical and chemical methods that can help digest the cotton–polyester matrix, making cotton more accessible for enzymatic degradation. In this study, NaOH pretreatment combined with mild hydrothermal treatment in the autoclave (121 °C for 15 min) of a 60/40 cotton/PET textile blend produced the best hydrolysis yield. Alkaline pretreatment has been widely used to depolymerize PET and decrease the crystallinity of cellulose [24,41]. The SEM micrographs clearly showed that the pretreated textile had an irregular and rough surface for more accessible enzymatic processing. Response surface methodology based on the central composite design is a promising tool to generate a highly significant quadratic polynomial to determine the appropriate concentrations of factors with substantial impact on hydrolysis yield of cotton-based textile waste hydrolysis. Jeihanipour and Taherzadeh [42] reported an almost complete conversion of cotton to glucose in alkaline-pretreated blue jean textiles. After 96 h of incubation, the textile was enzymatically hydrolyzed to 99.1% of the theoretical yield. The hydrolysis yield from NaOH pretreatment was positively correlated with the cotton content; a higher cotton content in the textile waste blend resulted in a higher hydrolysis yield. The hydrolysis of cellulose in cotton-based textile waste requires cellulase enzymes to work together [43,44]. The cellulose polymer is randomly cleaved at internal positions by endoglucanases (CMCase) into smaller sugars and oligomeric polysaccharides, which are further hydrolyzed by exoglucanases (FPase) to liberate soluble cellobiose or glucose. Finally, the β-glucosidases hydrolyze cellobiose to glucose.

From the results of this research, a new approach may be possible for addressing the environmental problems caused by the fashion industry, which are largely due to increased production and industrial demand. The fast fashion business model, which emphasizes mass production, variety, versatility, and affordability, contributes greatly to the amount and speed of waste generated [45]. Chemical treatment is considered a promising method for textile recycling, as it can break down textile waste into raw material monomers. Cotton waste from the textile industry has been identified as a suitable feedstock for producing animal feed, soil amendments, adsorbents, and construction materials [46]. Therefore, similar to recycled PET being used as value-added chemicals, this research could be an option for recycling textile waste by using recovered cotton cellulose and PET for other industries in the future.

## 4. Conclusions

This study successfully investigated potential pretreatment methods to modify textile waste for improved enzymatic hydrolysis. NaOH pretreatment combined with the mild hydrothermal process in autoclaved (121 °C for 15 min) was selected as it generated a high percentage hydrolysis yield with a cellulose content of 94.2 ± 0.57%. The results of the recyclability of NaOH solution indicated that this solution could be used for the pretreatment process of cotton/PET-based textile waste without replacement, and the structures and the obtained cellulose content of the pretreated samples are similar. This process significantly improved the hydrolysis of the 60/40 textile waste blend of cotton/PET and facilitated the fibers to be easily accessed to enzymatic hydrolysis. The regenerated cellulose was completely hydrolyzed under the optimized condition of 30 FPU/g of cellulase loading and 7% of substrate loading with a hydrolysis yield of 100% after 144 h of incubation. The results of this research suggest an alternative solution for the recovery and recycling of textile waste to alleviate the burden on the environment.

## Figures and Tables

**Figure 1 polymers-15-01964-f001:**
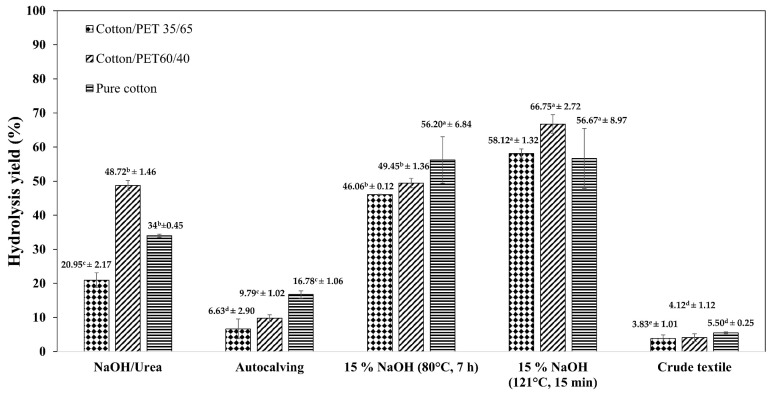
Hydrolysis yield of textile fibers by commercial cellulase under various pretreatment conditions. ^a,b,c,d,e^ mean with the different letters in the same pattern of the graph are significant at *p* ≤ 0.05.

**Figure 2 polymers-15-01964-f002:**
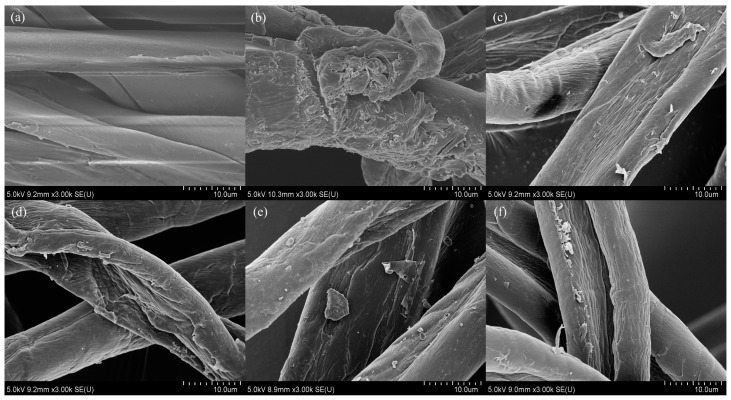
Scanning electron micrographs of textile waste (cotton/PET 60/40) before pretreatment (**a**) and after pretreatment with reusable NaOH with first cycle (**b**), second cycle (**c**), third cycle (**d**), fourth cycle (**e**), and fifth cycle (**f**) at a magnification of 3000×.

**Figure 3 polymers-15-01964-f003:**
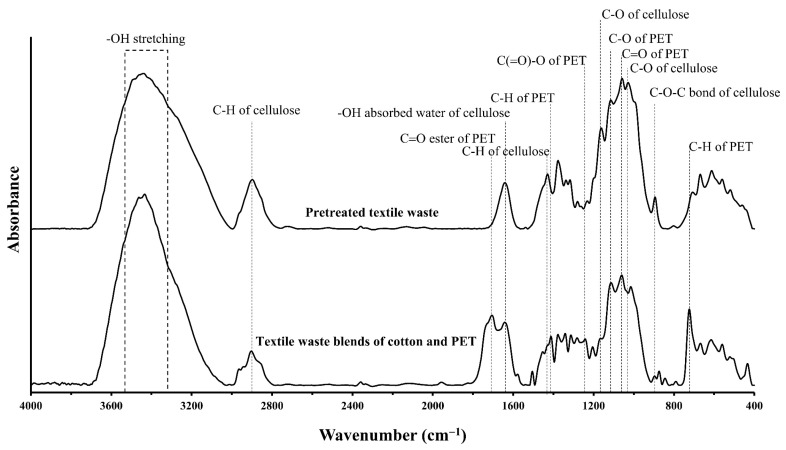
FTIR spectra of textile waste before and after pretreatment with 15% NaOH at 121 °C for 15 min in the range of 4000–400 cm^−1^ with resolution of 2 cm^−1^ and 32 scans.

**Figure 4 polymers-15-01964-f004:**
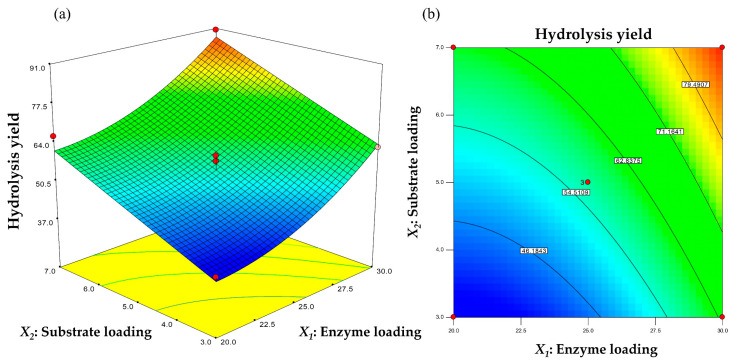
Response surface (**a**) and contour plots (**b**) of the combined effects between enzyme loading and substrate loading on hydrolysis yield.

**Figure 5 polymers-15-01964-f005:**
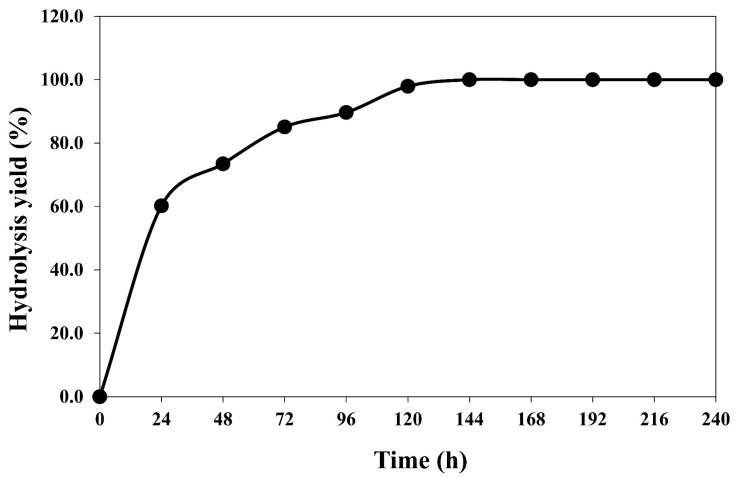
Time course of textile waste hydrolysis by commercial cellulase.

**Table 1 polymers-15-01964-t001:** Experimental design used in response surface methodology of two independent variables: enzyme loading ($X_{1}$) and substrate loading ($X_{2}$).

Run No.	Level		Actual Level		Hydrolysis Yield (%)	
	$X_{1}$	$X_{2}$	$X_{1}$	$X_{2}$	Observed	Predicted
1	−1	−1	20	3	39.21	37.85
2	1	−1	30	3	62.54	63.61
3	−1	1	20	7	66.34	61.37
4	1	1	30	7	90.36	87.81
5	−1.41	0	17.93	5	46.27	49.91
6	1.41	0	32.07	5	86.61	86.71
7	0	−1.41	25	2.17	40.68	40.12
8	0	1.41	25	7.83	69.31	73.76
9	0	0	25	5	57.59	56.78
10	0	0	25	5	53.17	56.78
11	0	0	25	5	59.59	56.78

**Table 2 polymers-15-01964-t002:** Analysis of variance for model regression representing hydrolysis yield of textile waste.

Source	Sum of Squares	DF	Mean Square	F	*p*-Value
Model	2707.56	5	541.51	30.14	0.001 *
$X_{1}$	1362.31	1	1362.31	75.82	0.0003 *
$X_{2}$	1138.8	1	1138.8	63.38	0.0005 *
$X_{1}X_{2}$	0.12	1	0.12	0.00641	0.9393
$X_{1}^{2}$	190	1	190	10.57	0.0227 *
$X_{2}^{2}$	0.037	1	0.037	0.00204	0.9658
Residual	89.84	5	17.97		
Lack of fit	68.26	3	22.75	2.11	0.3378
Total	2797.41	10			

* significance level = 95%; coefficient of determination (R^2^) = 0.968; Adjusted-R^2^ = 0.936.

## Data Availability

Data are contained within the article.

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
