# Peer review of "Optimization of Textile Waste Blends of Cotton and PET by Enzymatic Hydrolysis with Reusable Chemical Pretreatment"

_polymers, 2023, doi:10.3390/polym15081964_

Round 1

Reviewer 1 Report

Please find enclosed my comments on the manuscript number polymers-2277870 entitled "Optimization of textile waste blends of cotton and PET by enzymatic hydrolysis with reusable chemical pretreatment" submitted for publication in Polymers. The topic sounds interesting. However, it is needed to do a major revision before consider to publish in this journal.

1.       Add experimental procedure and condition of test in materials and methods section. Based on RSM test, add condition of each test.

2.       Discussion section should be added and more comparisons to other studies in Discussion section must be added.  Discussions of the results are not strong.

3.       Analytical Method is not clear. In this paper is there any statistical analysis? show statistical analysis on the figures.

Reviewer 2 Report

This paper is with good quality. The focus of the paper is on recycling
of textile and optimization method. Two small points to add:

1. Can the authors comment why using 35/65, 60/40 cotton/PET ratios for
the study, is there a specific reason for the DOE?
2. It seems to me the use of alkaline and acid to process waste textile
will consume more water, increase the cost, and can cause more water
pollution than the other methods. Can the authors comment on that?

Reviewer 3 Report

In the manuscript entitled "Optimization of textile waste blends of cotton and PET by enzymatic hydrolysis with reusable chemical pretreatment" the authors have described pretreatment methods for textile recycling, including autoclaving, freezing alkali/urea soaking, and alkaline pretreatment, with various cotton/PET blending ratios. They also optimized the hydrolysis of pretreated textile waste by cellulase using response surface methodology (RSM) based on central composite design (CCD); and found that the optimized conditions are 30 FPU/g of enzyme loading and 7% of solid loading, that resulted in a maximum observed value of hydrolysis yield at 89.7%, which is alomost similar to the predicted value of 87.8% after 96 h of incubation.

Before the accepatnce of the manuscript I have few queries and suggestions related to the manuscript.

1st the source of the textile waste and PET should be mentioned. As the authors mentioned in title they are working with textile waste the source should be mentioned specifically as the composition may vary from place to place and it may also affect the process. SImilarly all other chemicals the sources are important. 

The Figures should be prepared in high resolution with professional grpahical software such as origin, Graphpad Prism extra. The quality of the graph should be improved.

Authors should provide XPS along with analysis before and after treatment of the waste for better understanding of change in elemental composition.

In the introduction part the transition to cotton waste to PET should be smoother. Authors should establish some link with the above part. Or start the PET part in a separate paragraph. Authors should discuss more about recycling of PET by alkaline hydrolysis along with its contribution toward circular economy principal as authors are working with waste material. So that overall appeal of the manuscript can be improved. There are few important recent papers and reviews are missing related to PET depolymerization such as 

1) Accelerated Polyethylene Terephthalate (PET) Enzymatic Degradation by Room Temperature Alkali Pre-treatment for Reduced Polymer Crystallinity

2) Recent advances in chemical recycling of polyethylene terephthalate waste into value added products for sustainable coating solutions – hope vs. hype

3) Rapid hydrolysis of PET in high-concentration alcohol aqueous solution by pore formation and spontaneous separation of terephthalate.

The conclusion part should be more informative

Round 2

Reviewer 1 Report

This paper can be accepted. All of the comments are revised and edited.

Reviewer 3 Report

For conclusiveness of study XPS is needed, if more time is needed for the study authors should ask to editorial office for more time and then submit it.